# XLAND-100B: A LARGE-SCALE MULTI-TASK DATASET FOR IN-CONTEXT REINFORCEMENT LEARNING

**Alexander Nikulin**[*][†]   **Ilya Zisman**[*]   **Alexey Zemtsov**[*]   **Vladislav Kurenkov**
AIRI, MIPT              AIRI, Skoltech    NUST MISIS, T-Tech    AIRI, Innopolis University

## ABSTRACT

Following the success of the in-context learning paradigm in large-scale language and computer vision models, the recently emerging field of in-context reinforcement learning is experiencing a rapid growth. However, its development has been held back by the lack of challenging benchmarks, as all the experiments have been carried out in simple environments and on small-scale datasets. We present **XLand-100B**, a large-scale dataset for in-context reinforcement learning based on the XLand-MiniGrid environment, as a first step to alleviate this problem. It contains complete learning histories for nearly $30,000$ different tasks, covering 100B transitions and 2.5B episodes. It took $50,000$ GPU hours to collect the dataset, which is beyond the reach of most academic labs. Along with the dataset, we provide the utilities to reproduce or expand it even further. We also benchmark common in-context RL baselines and show that they struggle to generalize to novel and diverse tasks. With this substantial effort, we aim to democratize research in the rapidly growing field of in-context reinforcement learning and provide a solid foundation for further scaling.

## 1   INTRODUCTION

In-context learning, i.e. the ability to learn new tasks purely based on examples given in the context during inference and without any weight updates, was initially thought to be an emergent property of large language models, such as GPT-3 (Brown et al., 2020). However, it was quickly discovered that small transformers are also capable of in-context learning (Kirsch et al., 2022; Von Oswald et al., 2023), and even many non-transformer models have such abilities (Bhattamishra et al., 2023; Akyürek et al., 2024; Park et al., 2024; Grazzi et al., 2024; Vladymyrov et al., 2024; Tong & Pehlevan, 2024). More importantly, driven by properties of data, rather than the architecture (Chan et al., 2022; Gu et al., 2023), in-context learning is not specific to language modeling and has been found in other domains, e.g. image generation (Bai et al., 2023; Najdenkoska et al., 2023; Doveh et al., 2024; Tian et al., 2024).

However, despite the rapid adoption of the transformer architecture in reinforcement learning (RL) after the release of Decision Transformer (DT) (Chen et al., 2021; Hu et al., 2022; Agarwal et al., 2023; Li et al., 2023), models with in-context learning capabilities appeared only recently. This delay is caused by a number of reasons. Firstly, to transition from in-weights to in-context learning, a model should be trained on tens of thousands of unique tasks (Kirsch et al., 2022). Unfortunately, even the largest RL datasets currently contain only hundreds of tasks (Padalkar et al., 2023). Secondly, it was necessary to determine the right way to provide a context to a transformer and develop the data collection pipeline, which for many methods (Laskin et al., 2022; Lee et al., 2023; Shi et al., 2024) was different from what is commonly available in existing datasets (see Section 3).

Due to the lack of suitable datasets and the high cost of collecting data in existing environments, the recent wave of in-context RL research (Laskin et al., 2022; Lee et al., 2023; Norman & Clune, 2023; Kirsch et al., 2023; Sinii et al., 2023; Zisman et al., 2023) used environments with very simple task distributions, where it was feasible to collect datasets with hundreds of tasks. While these benchmarks are affordable, they are not suitable for the comparison of methods at scale on tasks

---

[*]Equal contribution.
[†]Correspondence to: nikulin@airi.net. Work done by dunnolab.ai, started at T-Tech.

Table 1: Comparison with other RL datasets.

| Data | # tasks | # transitions | # episodes | Size | Open-Source | Enables ICRL |
|---|---|---|---|---|---|---|
| **XLand-100B (ours)** | 28,876 | 100B | 2.5B | 320GB | ✓ | ✓ |
| JAT | 157 | 323M | N/A | 1TB | ✓ | ✗ |
| GATO | 596 | 1.5T | 63M | N/A | ✗ | ✗ |
| Open X-Embodiment | 527 | N/A | 2.4M | 9TB | ✓ | ✗ |
| AlphaStar Unplugged | 1 | 21B | 2.8M | N/A | ✗ | ? |
| NetHack | 1 | 3.5B | 110K | 97 GB | ✗ | ✗ |
| D4RL | 11 | N/A | 40 M | 4.8GB | ✓ | ✗ |
| V-D4RL | 4 | 2.4M | N/A | 17GB | ✓ | ✗ |
| D5RL | 50 | N/A | 2500 | N/A | ✗ | ✗ |
| RL Unplugged | 90 | 80M | N/A | 54.1TB | ✓ | ✗ |
| Procgen | 16 | 37M | N/A | 500GB | ✓ | ✗ |

with high diversity and difficulty, which is essential for real-world applications. Because of this, the development of in-context RL is currently hindered by these factors. We believe it is crucial to address these barriers, given the essential role of in-context learning in the path to foundation models and truly generalist agents (Team et al., 2021; 2023; Kirsch et al., 2023; Lu et al., 2024; Liu et al., 2024).

We release **XLand-100B**, a large-scale dataset for in-context RL based on the XLand-MiniGrid (Nikulin et al., 2023) environment. It contains complete learning histories for nearly $30,000$ different tasks, covering 100B transitions and 2.5B episodes. It took $50,000$ GPU hours to collect the dataset, which is beyond the reach of most academic labs. In contrast to most existing datasets for RL, our dataset is compatible with the most widely used in-context learning RL methods (see Section 3). With this substantial effort, we aim to democratize the research in the rapidly growing field of in-context RL and provide a solid foundation for further scaling.

Along with the main dataset, we provide a smaller and simpler version for faster experimentation, as well as utilities to reproduce or extend the datasets even further. We carefully describe the entire data collection procedure (see Section 4), providing all necessary details about the algorithm used for collection, filtering and relabelling with expert actions (see Section 4.2). We analyse the resulting dataset to ensure that we have met all the requirements for in-context RL (see Section 4.3). In addition, we conducted preliminary experiments with the common baselines on the collected datasets, showing there is still a lot of research needed to improve the in-context adaptation abilities on complex tasks (see Section 5).

## 2 BACKGROUND

### 2.1 IN-CONTEXT REINFORCEMENT LEARNING

Multiple methods for in-context RL have come out, each offering a different way of training and organising the context (Laskin et al., 2022; Lee et al., 2023; Mirchandani et al., 2023; Liu & Abbeel, 2023; Raparthy et al., 2023; Shi et al., 2024). We focus on Algorithm Distillation (AD) (Laskin et al., 2022) and Decision-Pretrained Transformer (DPT) (Lee et al., 2023), which we chose as the main methods for our work due to their simplicity and generality.

**Algorithm Distillation.** AD (Laskin et al., 2022) was one of the first to show that in-context learning was possible in RL, and captures the details of many other more recent methods (Mirchandani et al., 2023; Liu & Abbeel, 2023; Shi et al., 2024) while remaining very simple. It trains a transformer, or any other sequence model, to autoregressively predict next actions given the history of previous interactions, i.e. observations, actions and rewards. To transition from in-weights to in-context learning, it is essential that the **context should contain multiple episodes ordered by an increasing return**, which is different from the way it is done in DT-like methods (Chen et al., 2021; Janner et al., 2021; Lee et al., 2022).

**Decision-Pretrained Transformer.** DPT is an alternative approach inspired by the Bayesian inference approximation (Müller et al., 2021). Unlike AD, it trains a transformer to **predict the optimal action**

**for a query state given a random, task specific, context**. That is, the context that does not have to be ordered, but only has to contain transitions belonging to the same task. Thus, DPT requires access to optimal actions, but does not require a dataset of learning histories.

In addition, the theoretical analyses of AD and DPT methods (Lin et al., 2023; Wang et al., 2024) showed that they can implement near-optimal online RL algorithms such as Lin-UCB, Thompson sampling or even temporal difference (TD) methods solely during the forward pass.

## 2.2 XLand-MiniGrid

Starting from the seminal work of Wang et al. (2016); Duan et al. (2016); Finn et al. (2017) on meta-RL, much of the subsequent work (Zintgraf et al., 2019; Melo, 2022; Grigsby et al., 2023; Lu et al., 2024; Shala et al., 2024; Beck et al., 2024) has focused on environments that either have very simple task distributions, or have very small and limited distributions of hard tasks. This is because, to generalize in meta-RL, training needs to be performed on many different tasks, significantly increasing the cost and time required for experimentation. Recently, Nikulin et al. (2023) released XLand-MiniGrid, a GPU-accelerated environment and million-task benchmarks that significantly lowered the entry barrier for meta-RL research. We will describe it shortly here.



Figure 1: Visualization of a generic XLand-MiniGrid environment. Grid layout should be selected in advance, while the positions of the objects are randomized on each reset. For the dataset we use simpler layout with just one room, see Appendix M.

**Environment.** XLand-MiniGrid is a complete rewrite of MiniGrid (Chevalier-Boisvert et al., 2023) in JAX (Bradbury et al., 2018), incorporating a notion of rules and goals from XLand (Team et al., 2023). Leveraging JAX, it can run on a GPU or TPU accelerators at millions of steps per seconds. At its core, it is a goal-oriented grid-world environment with simple underlying dynamics, partial observability and sparse rewards. The action space is simple, consisting mainly of navigation and interaction with game objects, such as opening a door or picking and placing items. Observations are symbolic "images" encoding the agent surrounding as tile and color ID's. Rules are functions that can change the state of the environment based on some conditions, e.g. when two specific objects are places near each other, they both disappear and one new object is placed. Goals are similar, except they only validate some predefined conditions and do not change anything. Composing different rules and goals together we can create new tasks with varying reward and dynamics functions. For more detailed description we refer to Nikulin et al. (2023).

**Benchmarks.** Along with the environment itself, Nikulin et al. (2023) released a tool for the procedural generation of a vast number of unique tasks with varying levels of difficulty. Each task is represented by a binary tree, where the root is the goal to be achieved and rest of nodes define rules of the environment to be triggered in a recurring sequence. To standardize comparisons, four pre-sampled benchmarks with increasing diversity were provided: `trivial`, `small`, `medium`, `high`, each with one million unique tasks. For this work, we chose `medium` as a middle ground between yet unsolved `high` and less challenging `small` benchmarks. We also use `trivial` for smaller and simpler dataset version (see Section 4).

## 3 THE MISSING PIECE FOR IN-CONTEXT RL

In order to successfully train an in-context agent, training data shall meet certain criteria. To start with, the data should be comprised of actual learning histories (Laskin et al., 2022) or their approximations (Zisman et al., 2023), that contain enough exploration and exploitation phases of learning. Learning with just expert trajectories would not be sufficient for in-context ability to emerge, since an agent needs to know the history of policy improvement (Laskin et al., 2022; Kirsch et al., 2023). Another approach is to learn from optimal actions as proposed by Lee et al. (2023), but it is unclear how to access the optimal policies to get them. Besides, the data needs to contain thousands of different tasks to learn from (Kirsch et al., 2022). That is, for a simple task to find two squares on a $9 \times 9$ grid, an agent needs to see around 2000 different combinations of goals to start adapting for unseen

Table 2: Descriptive statistics of XLand datasets.

| Dataset | XLand-Trivial-20B | XLand-100B |
|---|---|---|
| Episodes | 868,805,556 | 2,500,152,898 |
| Transitions | 19,496,960,000 | 112,598,843,392 |
| History length | 60,928 | 121,856 |
| Num tasks | 10,000 | 28,876 |
| Max task rules | 0 | 9 |
| Observation shape | (5, 5) | (5, 5) |
| Num actions | 6 | 6 |
| Mean final return | 0.915 | 0.894 |
| Median final return | 0.948 | 0.925 |
| Median episode transitions | 22.45 | 57.75 |
| Disk size (compressed) | 60 GB | 326 GB |

locations (Laskin et al., 2022). Since such data was never collected and put into a single dataset, all current in-context RL practitioners were forced to generate data on their own, which inevitably added more complications in reproduction of the methods.

Besides, collecting thousands of different in-context episodes requires a considerable commitment, as training numerous RL agents is expensive in terms of time and resources. To that matter, the data used in current research is collected in very simplistic environments with straightforward goals, like reaching a specific target on a map (Laskin et al., 2022; Lee et al., 2023; Zisman et al., 2023) or to apply forces to actuators in order to walk a robot (Kirsch et al., 2023). This significantly slows down the pace of in-context RL research, as it is not only hard to test the applicability of proposed methods, but also yet unfeasible to determine the scaling laws in these environments.

To provide a complete picture for the reader, we briefly discuss the existing datasets and highlight why they are unfit for training in-context RL agents. For simplicity, we categorize them into two groups: classical datasets designed for offline-RL and datasets collected for large-scale supervised learning. Note that this categorization is fuzzy in nature and serves only for better understanding of the current structure in RL data.

**Offline RL Datasets.** The datasets in this category can be considered classical, as some of them exist for more than four years now (Fu et al., 2020). They were initially proposed for offline RL, containing simple tasks with a flat structure, e.g. perform locomotion with different robots (Lu et al., 2023) or path finding in a maze. Some of them also contain data from robotic manipulators (Fu et al., 2020; Rafailov et al., 2023), or even Atari frames (Gulcehre et al., 2021). Other datasets collect data for more sophisticated environments, such as NetHack Learning Environment (Hambro et al., 2023; Kurenkov et al., 2024) or ProcGen (Cobbe et al., 2019; Mediratta et al., 2024). However, the aforementioned datasets offer $< 100$ different tasks with a fixed policy (except for the -replay datasets, which have limited coverage of various policies). This limitation makes it difficult for in-context RL to emerge from such data. To overcome this pitfall, we collected almost $30,000$ tasks with a deep ruleset structure, that are a challenging problem to solve.

**Large-Scale Supervised Pretraining.** Recent progress in generalist agents, which can solve a multitude of environments, has been made possible thanks to large datasets. GATO dataset (Reed et al., 2022), however not being released to the public, consists of 1.5 trillion transitions along with 596 tasks, which makes it one of the largest dataset in RL. The open-sourced analogue, the JAT dataset (Gallouédec et al., 2024), is smaller in size with 157 tasks and 300 million transitions, but it provides comparable performance on most of the benchmarks. Both datasets contain expert RL demonstrations from BabyAI (Hui et al., 2020), Atari games (Bellemare et al., 2013), Meta-World (Yu et al., 2020) and more.

Another large dataset, Open X-Embodiment (Padalkar et al., 2023), is a combination of more than 60 datasets from different robotics research labs. It consists of 527 different tasks in robotics with the demonstrations from mostly human experts. Despite the large quantity of transitions in these datasets, they do not contain learning histories with improving policies, making their application for in-context

RL quite challenging. On the contrary, our XLand-100B dataset consists of 100 billions of transitions of RL agents' learning histories, making it possible for in-context abilities to emerge.

The only potentially applicable dataset to use for in-context RL is AlphaStar Unplugged (Mathieu et al., 2023). Although the authors did not initially plan to collect a suitable dataset, the data can be sorted by players' MMR (analogous to Elo rating). This sorting can be considered a steady policy improvement, thus enabling the in-context RL ability. For more details on the datasets, refer to Table 1.

# 4   XLAND-100B DATASET

We present **XLand-100B**, a large dataset for in-context RL, and its smaller and simpler version **XLand-Trivial-20B** for faster experimentation. Together they contain about 3.5B episodes, 130B transitions and 40,000 unique tasks (see Table 2 for detailed statistics). Datasets are hosted on public S3 bucket and will be freely available for everyone under CC BY-SA 4.0 licence. Next, we describe the data format, collection and evaluation.

## 4.1   DATA FORMAT

**Storage format.** We chose to store the datasets in HDF5[1] file format based on its popularity and convenience. It allows to work with large amounts of structured data without loading it to memory, store arbitrary metadata, and customise compression and chunk size to maximize the sampling throughput. We used gzip compression with default compression strength of 6, which reduced dataset size from almost 5TB+ to just ∼600GB for our main dataset. Using a little trick described later, we were able to reduce the size even more to just 326 GB (see Table 2). However, a naive use of compression can dramatically increase batch sampling time and slow overall training time down. We tuned HDF5 cache chunk size specifically to maximize sampling throughput for large sequence lengths. After tuning, we achieved a fourfold speedup over the naive compression, and were only two times slower compared to no compression. Given that we reduced the dataset size by a factor of 15, we see this as a good trade-off, which increases the overall affordability of the dataset. See Appendix C for throughput benchmarks.

**Data and metadata format.** We collect complete learning histories, i.e. for each history we store all observations, actions, rewards and dones encountered during agent training in separate HDF5 groups with unique IDs per history (see Appendix B for more details). To be compatible with DPT-like methods (Lee et al., 2023), we also store expert actions for each transition (see Section 4.2). Unlike popular formats such as RLDS[2] and Minari[3], we store transitions as one sequential array per history per modality. The rationale here is that under compression, it is much cheaper to sample slices of long episodes in sequence rather than sampling across different groups.

We also store observations efficiently to further reduce dataset size. Instead of storing two channels for tile and color, we map their indexes into Cartesian product of colors and tiles, halving the storage size. They can be decoded easily during sampling without any overhead with `divmod` function. In addition, for each history we store the XLand-MiniGrid environment ID, benchmark ID and ruleset ID, which can be used later to filter the dataset, e.g. based on the complexity of the tasks, split into train and test or set up the environment for evaluation.

## 4.2   DATA COLLECTION

At a high level, data collection was organized into three stages, namely multi-task task-conditioned pre-training; single-task fine-tuning to collect learning histories; and finally post-processing and filtering. Although we used highly optimised GPU-accelerated implementations of the base RL algorithm and environment, it still took 50, 000 GPU hours to collect the full dataset. Collecting a dataset of this size for any other environment suitable for in-context RL, e.g. Meta-World (Yu et al., 2020), would take much longer, which is unlikely to be feasible for most practitioners (Nikulin et al.,

---

[1] https://github.com/HDFGroup/hdf5
[2] https://github.com/google-research/rlds
[3] https://github.com/Farama-Foundation/Minari

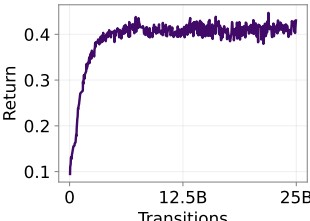 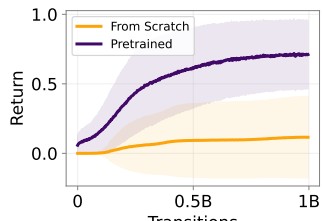 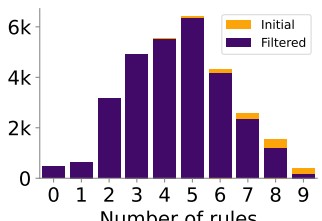

Figure 2: Evaluation return for multi-task goal-conditioned reccurent PPO pretraining on 65k tasks. Pretrained agent was further used a starting point for single-task finetuning during dataset collection.

Figure 3: Single-task evaluation curves on 36 hard tasks for policies trained from scratch or fine-tuned from multi-task pretrained checkpoints. See Appendix D for curves on tasks of all difficulty.

Figure 4: Distribution of the tasks by difficulty sampled initially and in the resulting dataset. To ensure the quality, we filtered tasks where the final return was below 0.3 or the data was corrupted due to some errors during training.

2023). Next, we describe the collection process, including the selection of the base RL algorithm and all subsequent steps. We provide exact hyperparameters for each stage in Appendix O.

**Base algorithm.** For our datasets we chose PPO (Schulman et al., 2017), due its high scalability and compatibility with massively parallel environments. We ported the implementation from recurrent PPO provided by (Nikulin et al., 2023), customizing it to meet our needs. We added callbacks for saving transitions during training and extended agent architecture to take ruleset encoding as an optional condition for pre-training. We used GRU (Cho et al., 2014) for memory, as it showed satisfactory performance on preliminary experiments. Since the base algorithm was implemented in JAX (Bradbury et al., 2018), we were able to just-in-time compile the entire training loop, achieving 1M steps per second during training on one GPU with mixed precision enabled. As PPO is not the most sample-efficient algorithm, it was still necessary to train it on billions of transitions. Fortunately, as we trained it on thousands of environments in parallel, learning history for each particular environment is quite short, i.e. around 120k transitions.

**Pretraining.** For our main XLand-100B dataset we uniformly sampled tasks from `medium-1m` benchmark from XLand-MiniGrid. It contains tasks of various difficulty, ranging from zero to nine rules. Unfortunately, on many hard tasks our base algorithm could not manage to converge in the time budget allocated for a single training run. On the hardest tasks, it was not even possible to get a non-zero reward at all, due to the exploration challenge that such tasks poses. In order to speed up convergence and exploration on harder tasks, we pre-train an agent in a multi-task task-conditioned manner and also use simpler grid layout with just one room (see Appendix M). We expose the ruleset specification, which is usually hidden from the agent, and encode the goal and rules via embeddings, concatenating resulting encodings and passing it as an additional input to the agent. After that, we train the agent on 65k tasks simultaneously for 25B transitions. As Figure 2 shows, such an agent learns to generalize zero-shot on new tasks quite well, although we do not aim to push it to the limit, as zero-shot generalization does not produce a smooth learning history during fine-tuning. We skip this stage for XLand-Trivial-20B dataset due to the simplicity of the tasks in `trivial-1m` benchmark.

**Finetuning.** This is a key stage in the data collection process, during which we finetune a pretrained agent while recording the transitions encountered into the dataset. We finetune the agent using 8192 parallel environments for 1B transitions on 30k uniformly sampled tasks from `medium-1m` benchmark. We mask out the task-conditioning encoding to prevent zero-shot generalization. We record transitions only from first 32 parallel environments. This way, we can keep the size of the dataset manageable, still leaving the possibility to study scaling laws and generalization in a controlled manner. For example, we can train AD (Laskin et al., 2022) on all 30k tasks using one history per task or on $\sim$ 900 tasks using all histories per task to equalize the number of training tokens. For the XLand-Trivial-20B dataset, instead of fine-tuning, we train the agent from scratch on 10k uniformly sampled tasks from the `trivial-1m` benchmark, keeping all other hyperparameters the same. In the Figure 3 we show the effect of finetuning on hard tasks (with more than seven rules) compared to training from scratch. It can be seen that we are able to show strong performance even on the hardest

tasks, increasing the diversity and coverage of the resulting dataset. For the same results on tasks of all levels of difficulty, see Appendix D.

**Postprocessing.** After fine-tuning, it was necessary to additionally label the transitions with the expert actions to support DPT-like methods (Lee et al., 2023). To do this, we walk through the entire learning history with the final policy, starting from the initial hidden state for the RNN. We evaluate the validity of such a labelling scheme later in the Section 4.3. Finally, all individual learning histories from different tasks were combined into one large dataset. To ensure quality, we filtered out any task with a final return below 0.3 as an unrepresentative learning history. There were some failures, such as GPU crashes, which are inevitable during large-scale training. So any runs with corrupted data were also filtered out. In total, we filtered out about 1k tasks, leaving almost 29k tasks in the final dataset. We provide detailed statistics for each dataset in Table 2 and the final distribution of tasks by number of rules in Figure 4.

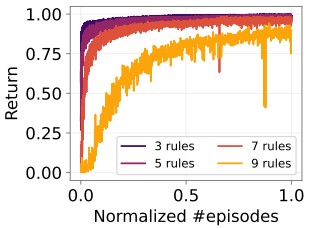

Figure 5: Learning histories for the XLand-100B dataset separated by number of rules. For visual clarity, we show only a sample of the possible number of rules and normalize the number of episodes, as they may vary considerably.

### 4.3 DATA EVALUATION

In this section we validate that the resulting XLand-100B dataset actually fulfils the two most important requirements for in-context RL, namely it contains learning histories with distinct policy improvement pattern and has expert actions for each transition (see Section 3 for a discussion). We provide analogous results for XLand-Trivial-20B in the Appendix E.

**Improvement history.** In the Figure 5 we show the averaged return from the learning histories separated by the number of rules. In order to better show the speed of learning on the same scale, we have normalized the x-axis for each level of difficulty, as the number of episodes can vary greatly (as it takes more time to solve complex tasks). One can see that the dataset provides a whole range of learning speeds, from very fast on easy problems to much slower on the hardest, which may be important for methods based on AD (Zisman et al., 2023; Shi et al., 2024). For a learning history averaged over full dataset see Appendix E.

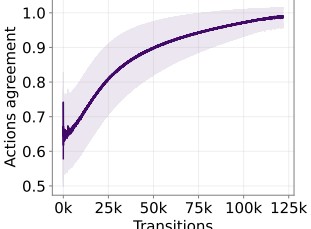

Figure 6: Agreement between actions predicted by the expert and the actual actions in the learning history. We use final PPO policy as an expert for actions labeling. As can be seen agreement is growing towards the end of learning history.

**Expert actions relabeling.** In contrast to AD, which predicts next actions from the trajectory itself, DPT-like methods require access to optimal actions on each transition for prediction. However, for the most nontrivial or real-world problems, obtaining true optimal actions in large numbers is unlikely to be possible. Recently, Lin et al. (2023) introduced approximate DPT, scheme where expert actions are estimated from the entire history by some algorithm. We implemented this scheme for lack of evident alternatives. However, we had to make sure that such a labeling is adequate in our case and the expert at the end predicts actions close to what the policy did in reality near the end of training. This is not obvious, as during the labeling it can diverge into out-of-distribution hidden states for RNN. In the Figure 6 we show that on XLand-100B the agreement between the predicted actions by the expert and the actual actions increases closer to the end of the learning history, meaning that the expert does not diverge during relabeling.

## 5 EXPERIMENTS

In this section, we investigate whether our datasets can enable an in-context RL ability. Additionally, we demonstrate how well current in-context algorithms perform across different task complexities and outline their current limitations. We take AD (Laskin et al., 2022) and DPT (Lee et al., 2023) for our experiments, the exact implementations details are in Appendix F and Appendix G. Both methods were trained on `XLand-Trivial-20B` and `XLand-100B` with {512, 1024, 2048, 4096} and {1024, 2048, 4096} context lengths respectively. We do not include a 512 context length for the

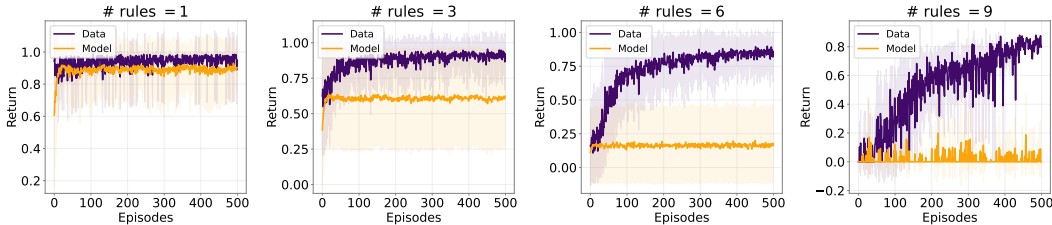

Figure 7: Comparison of the learning histories in `-100B` dataset vs. AD performance on the same *training* tasks. AD is able to solve simple tasks, however its performance degrades as the rulesets get deeper. The context length of the model is 1024. The evaluation parameters, except training tasks, are the same as in Figure 8.

`-100B` dataset as we consider it too short, given the nearly 3x increase in median episode length between the two datasets. For evaluation, we run three models on 1024 unseen tasks for 500 episodes.

**AD.** Algorithm Distillation shows an emergence of in-context ability during training on both datasets. Figure 8 demonstrates the performance of the method for different context lengths. On `-Trivial-20B` dataset it is able to show a stable policy improvement from about 0.28 to 0.4 during the evaluation. For `-100B` the performance is similar, but the pace of improvement is faster. We hypothesise that it happens due to wider data-coverage, since the agent sees more complex tasks and is able to learn faster from them.

To further examine the performance on the `-100B` dataset, we evaluate AD on the training tasks from the dataset and separate the performance based on the complexity of the tasks. The complexity is defined by the number of rules an agent needs to trigger before the successful completion. As shown in Figure 7, AD is able to demonstrate in-context abilities on simple tasks, but it struggles with more complex ones. We speculate that our results are not final, as AD is not yet fully capable of learning to solve the training tasks. We believe there is a need for further research to discover new and more sample-efficient architectures capable of solving the more complex rulesets of our dataset.

**DPT.** Decision-Pretrained Transformer (Lee et al., 2023) is another method that exhibit in-context RL capabilities. However, in our experiments we were unable to train it so that these abilities emerge. Figure 18 in Appendix K demonstrates the lack of performance even on the simplest trivial tasks. We believe this is closely connected to the inability of DPT to reason in POMDP environments. For a detailed investigation, we refer the reader to Appendix H.

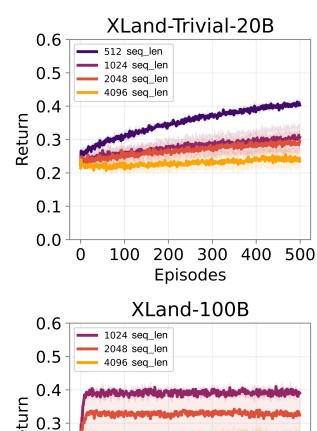

Figure 8: AD performance on our datasets for different sequence lengths. Both datasets lead to the emergence of in-context ability. We report the average return on 1024 unseen tasks across 3 seeds.

## 6 LIMITATIONS AND FUTURE WORK

There are several limitations to our work, some of which we hope to address in future releases. Despite the size and diversity of the datasets provided in terms of tasks, we do not provide diversity in terms of the domains, as all tasks share the same observation and action spaces. In addition, the tasks also share an overall latent structure, i.e. it is always a form of binary tree. This can be addressed with more diverse benchmark generators in the XLand-MiniGrid library (Nikulin et al., 2023). All learning histories were collected on grids with only one room, which may limit the transfer to the harder layouts with multiple rooms containing doors. Finally, the effect of fine-tuning from pre-trained checkpoints is underexplored and could potentially hurt performance, as there are many learning histories that start from a high reward. We hope to improve the RL baseline for data collection to avoid the need for multi-task pre-training in the future.

ETHICS STATEMENT

**Legal Compliance and Licensing.** The dataset will be released under the CC BY-SA 4.0 license, and the accompanying code will be available under the Apache 2.0 license. We have ensured that our data and code releases comply with all relevant legal and ethical guidelines.

**Data Privacy and Security.** The dataset does not include any human subjects or personally identifiable information. All data are generated by artificial agents interacting within a simulated environment. Consequently, there are no privacy or security concerns associated with the dataset.

**Potential Misuse and Harmful Applications.** The dataset and methods are intended to advance research in in-context reinforcement learning. We do not anticipate any immediate risks of misuse or harmful applications arising from our work. Nonetheless, we encourage users to apply the dataset responsibly and adhere to ethical standards in their research.

**Environmental Impact.** Collecting the dataset required substantial computational resources, totaling approximately 50,000 GPU hours. We acknowledge the environmental footprint associated with large-scale computations. By openly releasing this dataset, we aim to minimize redundant data collection efforts by other researchers, potentially reducing the overall environmental impact in this field.

REPRODUCIBILITY STATEMENT

To ensure reproducibility we share the important details throughout the paper and appendix. We explain how we pretrain, finetune models for collecting data and the post-processing step in Section 4.2, as well as discussing the dataset details (size, compression ratios, etc.) in Appendix B and Appendix C. For experiments section, we discuss the details of baselines implementation in Appendix F and Appendix G. We provide hyperparameters for collecting the dataset and for baselines in Appendix O. We also release the codebase with tools for creating and expanding the dataset in the following repository: xland-minigrid-datasets.

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

## A    DOWNLOADING THE DATASETS

Both `XLand-100B` and `XLand-Trivial-20B` datasets hosted on public S3 bucket and are freely available for everyone under CC BY-SA 4.0 Licence. We advise starting with Trivial dataset for debugging due to smaller size and faster downloading time. Datasets can be downloaded with the curl (or any other similar) utility.

```
# XLand-Trivial-20B, approx 60GB size
curl -L -o xland-trivial-20b.hdf5 https://tinyurl.com/trivial-10k

# XLand-100B, approx 325GB size
curl -L -o xland-100b.hdf5 https://tinyurl.com/medium-30k
```

## B    WHAT IS INSIDE DATASET?

Both `-Trivial` and `-100B` dataset are HDF5 files holding the same structure. The dataset is grouped the following way:

```
data["{key_id}/{entity_name}"][learning_history_id]
```

where `key_id` is an ordinal number of a task in dataset, `learning_history_id` is a learning history number from 0 to 32 and `entity_name` is one of the names mentioned in Table 3.

**NB!** Do not confuse `key_id` with the task ID, which should be accessed via

```
data["{key_id}"].attrs["ruleset-id"]
```

Table 3: Data description

| Name | Type | Shape | Description |
|------|------|-------|-------------|
| states | np.uint8 | (5, 5) | $s_t$, colors and tiles from agent's POV |
| actions | np.uint8 | scalar | $a_t$, from 0 to NUM_ACTIONS |
| rewards | np.float16 | scalar | $r_t$, which agent recieved at timestep $t$ |
| dones | np.bool | scalar | $d_t$, terminated or truncated episode flag |
| expert_actions | np.uint8 | scalar | same as $a_t$ but from a generating policy |

## C    COMPRESSION CHUNK SIZE TUNING

Table 4: Throughput with PyTorch dataloader with different HDF5 compression chunk size settings. We used 2048 sequence length, 64 batch size and 8 workers. Chunking was applied along the learning history dimension.

| Compression | Chunk size | Throughput |
|-------------|-----------|-----------|
| None | None | 1,549,619 |
| gzip | None | 173,895 |
| gzip | 256 | 423,513 |
| gzip | 512 | 549,397 |
| gzip | 1024 | 666,152 |
| gzip | 2048 | 768,706 |
| gzip | 4096 | 749,851 |
| gzip | 8192 | 737,646 |

## D ADDITIONAL FIGURES OF DATA COLLECTION

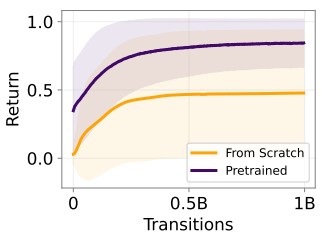

Figure 9: Single-task evaluation curves on 256 tasks for policies trained from scratch or fine-tuned from multi-task pretrained checkpoints.

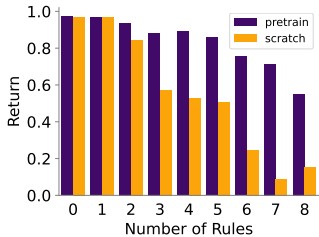

Figure 10: Final return by number of rules on 256 tasks for policies trained from scratch or fine-tuned from multi-taks pretrained checkpoints.

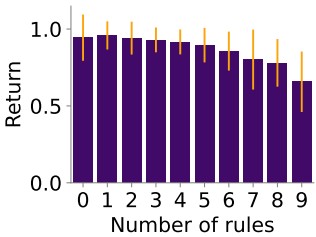

Figure 11: Final return by number of rules in the final XLand-100B dataset after post-processing.

## E ADDITIONAL FIGURES OF DATA EVALUATION

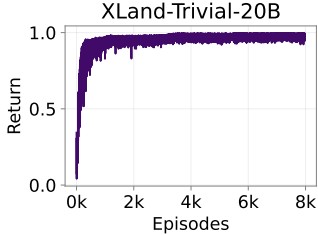

Figure 12: Return averaged over all learning histories in the final XLand-Trivial-20B dataset.

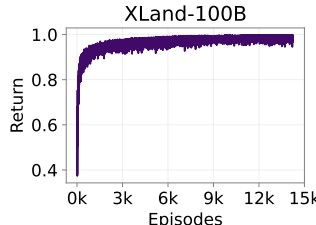

Figure 13: Return averaged over all learning histories in the final XLand-100B dataset.

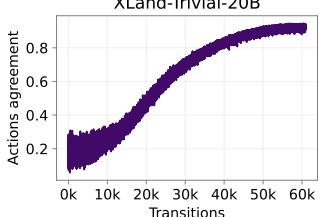

Figure 14: Agreement between actions predicted by the expert and the actual actions in the learning history. We use final PPO policy as an expert for actions labeling.

# F  AD IMPLEMENTATION

Table 5: Time for training and evaluation.

| sequence length | 1024 | 2048 | 4096 |
|---|---|---|---|
| train | 15 hrs | 10 hrs | 11.5 hrs |
| eval | 20 min | 30 min | 50 min |

We implement AD following the original paper of Laskin et al. (2022). To optimize the speed of training and inference we use FlashAttention-2 (Dao, 2024) with KV-caching and ALiBi positional embeddings (Press et al., 2022). We also concatenate observations, actions and rewards along the embedding dimensions, as it reduces the context size by the factor of three. We also use DeepSpeed (Rasley et al., 2020) to enable distributed training. The total number of parameters of our model is 25 M.

The approximate time of training for single epoch on a `-100B` dataset and evaluation on 1024 tasks on 8 H100 GPUs is shown in the Table 5. The computations were done on an internal cluster.

The hyperparameters was copied from (Laskin et al., 2022) except for the size of the network, it was scaled up to 25 M. The exact hyperparameters can be found in Table 8.

We also show training logs for both datasets. Trivial: wandb; medium: wandb

# G  DPT IMPLEMENTATION

Our implementation is based on the original one (Lee et al., 2023). Compared to AD implementation (Laskin et al., 2022), during training phase, the model context is generated with decorrelated dataset transitions to increase data diversity and model robustness: given a query observation and a respective expert action for it, an in-context dataset is provided by random interactions within the same ruleset. During evaluation phase, multi-episodic contextual buffer consists only of previous episodes and updates after the current one ends. The intuition behind this approach is the observed policy during any given episode is fixed, so it is a lot easier to analyze this policy than a dynamically changing one while it is executing.

Both AD and DPT shares the same observation encoder and transformer block, except there is no positional encoding in DPT, as stated in (Lee et al., 2023).

The training consisted of 3 epochs due to computational and time limitations as 1 epoch approximately lasted 12 hours, while the evaluation on 1024 rulesets on 500 episodes could take from 5 to 21 hours, depending on the model's sequence length. All experiments ran on 8 A100 GPUs. The computations were done on an internal cluster.

The hyperparameters was copied from (Lee et al., 2023) except for the size of the network, it was made up to 25 M, and sequence length, it was increased due to complexity of the tasks. The exact hyperparameters can be found in Table 7.

We also show DPT training logs. Trivial: wandb; medium: wandb

# H  ON DPT LIMITATIONS IN POMDP

We additionally demonstrate the inability of DPT to learn in-context in Partially Observable MDP (POMDP) on the example of a toy Dark Key-To-Door environment (Laskin et al., 2022). The agent is required to find an invisible key and then open an invisible door. The reward of 1 is given when the agent first reaches the key and then the door. Note that the door cannot be reached until the key is found. This way Key-To-Door can be considered a POMDP. However, the environment can be reformulated as an MDP by providing additional boolean indicator of reaching a key in addition to the agent's position. This way, algorithms that work only with MDPs are able to solve this environment.

Based on this fact, we learn two different Q-tables for both environments: with and without the key indicator. The learning histories of Q-Learning algorithm are stored together with optimal actions computed via the oracle.

For clarity, we call DPT training and evaluation Markovian when "reached key" indicator is provided for every state. We trained DPT on Key-To-Door for $150,000$ updates in Markovian and non-Markovian setups to show the difference in performance. As it can be seen in Figure 15, in the Markovian case the model converges to the optimal return, finding both a key and a door. In the latter case, the model reaches a plateau reward of $1$ which means it finds a key. As we empirically observe, without the indicator DPT reaches only a suboptimal return. We believe it happens due to DPT inability to reason whether a key was already found from the random context. Without this knowledge, it is impossible for the agent to know whether it should search for the key or for the door.

We also show logs for Key-To-Door Experiments: Markovian: wandb; non-Markovian: wandb

Table 6: Q-Learning Hyperparameters

| Hyperparameter | Value |
| --- | --- |
| Num. Train Goals | 2424 |
| Num. Histories | 5000 |
| Num. Updates | 50,000 |
| Learning Rate | 3e-4 |

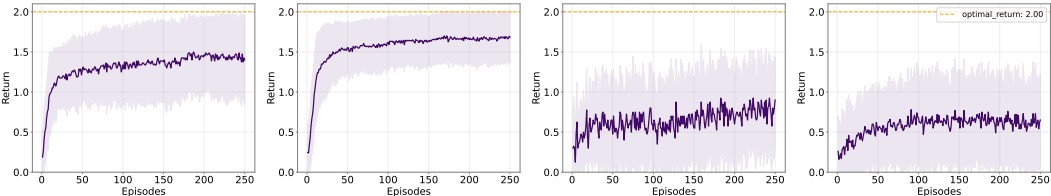

Figure 15: The performance of DPT on Key-To-Door environment. From left to right: first two plots indicate returns on 20 train and 50 test goals, respectively, for Key-To-Door as MDP by providing additional indicator of reaching the key to the model. Second two plots indicate returns with the model that has no access to the fact of reaching a key. The results are averaged across four seeds

## I  DETAILS OF IN-CONTEXT EVALUATION

The evaluation process resembles the standard in-weight learning, with the key difference being that the learning itself happens during evaluation. When interacting with the environment, the agent populates the Transformer's context with the latest observations, actions and rewards. At the start of the evaluation, the context is empty. Note that the agent's context is cross-episodic, which makes it possible for the agent to access transitions in previous episodes. We report the cumulative reward that the agent achieved at the end of each episode. We run evaluation for each model for 500 episodes, reporting mean return across 1024 unseen tasks with standard deviation across 3 seeds. We claim that the in-context emerged when the mean return rises up until some level. This means the agent improves its policy from episode to episode, learning how to solve a task.

# J    ADDITIONAL FIGURES OF AD PERFORMANCE

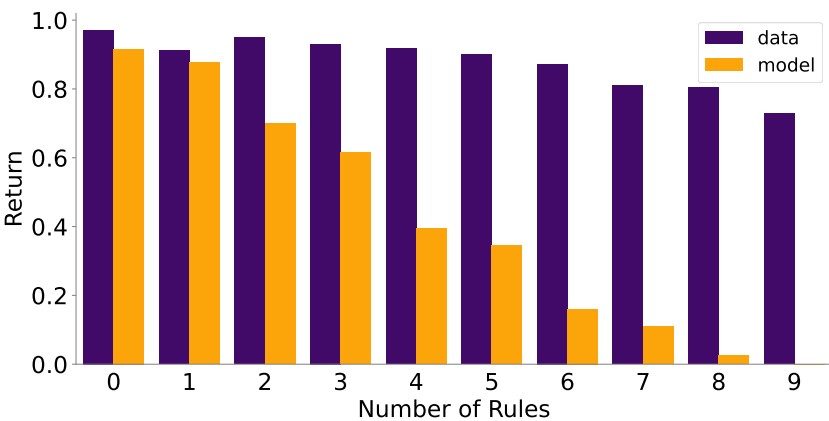

Figure 16: AD performance on different complexities of rulesets. AD is evaluated on 1024 training tasks from `-100B`. Sequence length is 1024.

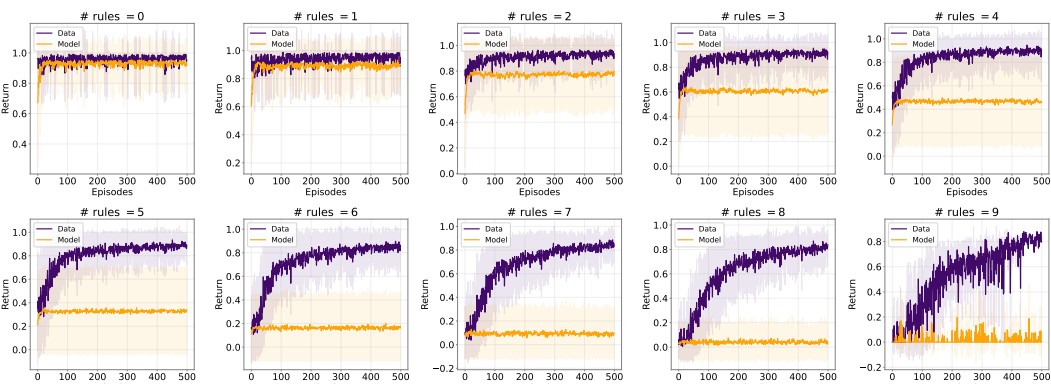

Figure 17: AD's performance on different task complexities, a full variant of Figure 7.

# K    ADDITIONAL FIGURES OF DPT PERFORMANCE

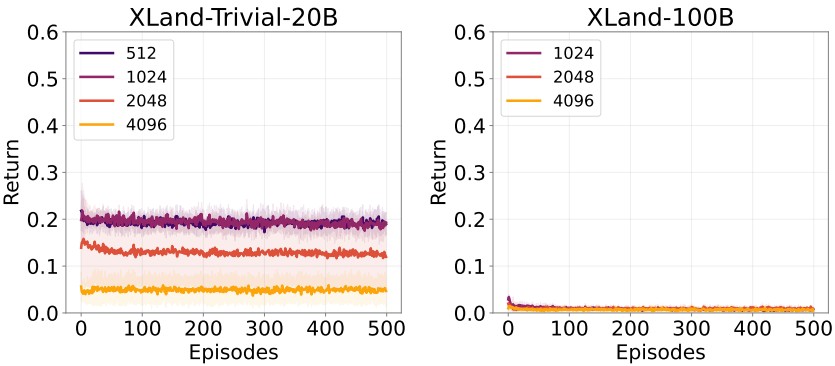

Figure 18: DPT performance on both datasets. Evaluation parameters are the same as in Figure 8.

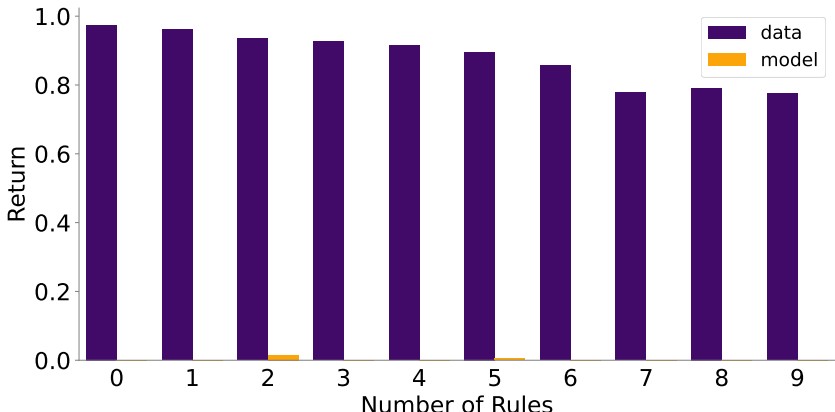

Figure 19: DPT performance on different complexities of rulesets. DPT is evaluated on 1024 training tasks from `-100B`. Sequence length is 1024.

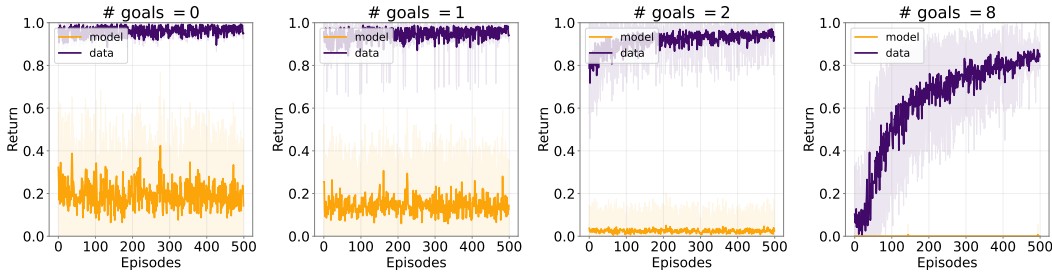

Figure 20: Comparison of the learning histories in `-100B` dataset vs. DPT performance on the same *training* tasks. DPT is not able to solve simple tasks and there is no observation in-context learning emerges, model's performance also degrades as the rulesets get harder. The context length of the model is 1024. The evaluation parameters, except training tasks, are the same as in Figure 8.

## L    RULESETS VIZUALIZATION

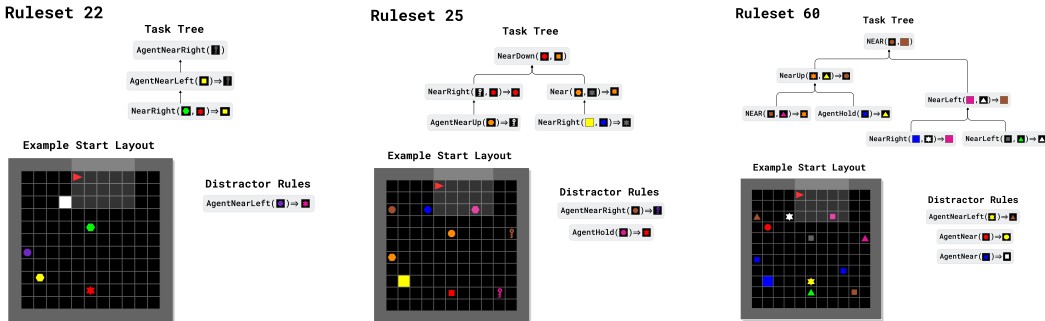

Figure 21: Rulesets with increasing difficulty (i.e. with 3, 5, and 9 rules respectively) sampled from medium-1m benchmark. They can be accessed by ID's via the XLand-MiniGrid package.

## M    GRID LAYOUTS USED FOR DATASET COLLECTION

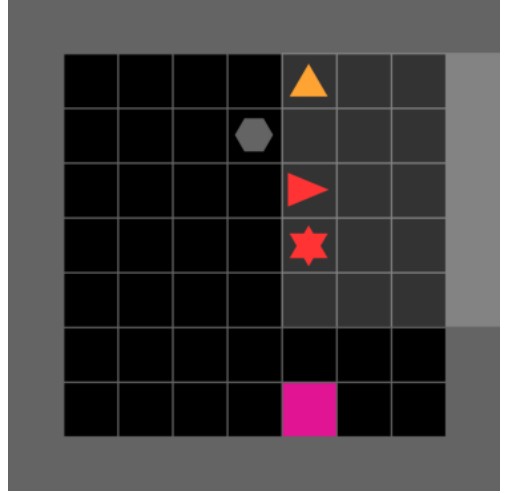

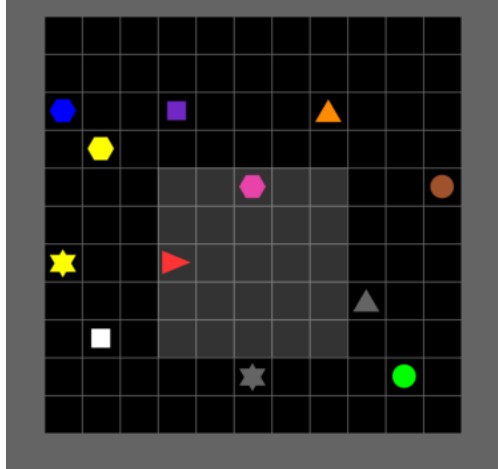

Figure 22: Layout used for XLand-Trivial-20B. Corresponds to XLand-MiniGrid-R1-9x9 environment from the XLand-MiniGrid suite. Game objects are task dependent.

Figure 23: Layout used for XLand-100B. Corresponds to XLand-MiniGrid-R1-13x13 environment from the XLand-MiniGrid suite. Game objects are task dependent.

## N    ADDITIONAL EXPERIMENTS ON NUMBER OF TASKS

We run additional experiments with different number of tasks to show how it can affect the performance. Experiments were done with the sequence length of 2048. To make the number of gradient updates closer to the original experiments, we progressively increased the number of epochs for each smaller subset. Note that the return increases from 100 goals to 3k goals, indicating the importance of scaling the number of goals in the dataset. Running the AD for a single epoch on full goals dataset is clearly not enough, however, a large scale training is too computationally expensive and it is left for future research.

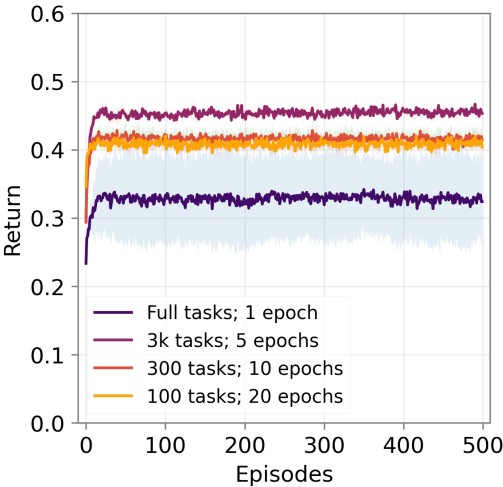

Figure 24: Training AD for different number of goals.

## O HYPERPARAMETERS

### Table 7: DPT Hyperparameters

(a) DPT Hyperparameters for datasets

| Hyperparameter | Value |
| --- | --- |
| Embedding Dim. | 64 |
| Number of Layers | 8 |
| Number of Heads | 8 |
| Feedforward Dim. | 256 |
| Layernorm Placement | Pre Norm |
| Embedding Dropout Rate | 0.1 |
| Batch size | [512, 256, 128] |
| Sequence Length | [1024, 2048, 4096] |
| Optimizer | Adam |
| Betas | (0.9, 0.99) |
| Learning Rate | 1e-3 |
| Learning Rate Schedule | Cosine Decay |
| Warmup Ratio | 0.05 |
| # Parameters | 25 M |

(b) DPT Hyperparameters Key-to-Door

| Hyperparameter | Value |
| --- | --- |
| Embedding Dim. | 64 |
| Number of Layers | 4 |
| Number of Heads | 4 |
| Feedforward Dim. | 64 |
| Layernorm Placement | Pre Norm |
| Residual Dropout | 0.5 |
| Sequence Length | [50, 100, 250, 350, 500] |
| Batch size | 128 |
| Optimizer | Adam |
| Betas | (0.9, 0.99) |
| Learning Rate | 1e-3 |
| Label Smoothing | 0.3 |
| Learning Rate Schedule | Cosine Decay |
| Warmup Ratio | 0.05 |
| # Parameters | 200 K |

### Table 8: AD Hyperparameters

| Hyperparameter | Value |
| --- | --- |
| Embedding Dim. | 64 |
| Number of Layers | 8 |
| Number of Heads | 8 |
| Feedforward Dim. | 512 |
| Layernorm Placement | Pre-norm |
| Embedding Dropout | 0.1 |
| Batch size | [256, 128, 64] |
| Sequence Length | [1024, 2048, 4096] |
| Optimizer | Adam |
| Betas | (0.9, 0.99) |
| Learning Rate | 1e-3 |
| Learning Rate Schedule | CosineLR |
| Warmup Steps | 500 |
| # Parameters | 25 M |

Table 9: PPO hyperparameters used in multi-task pre-training from Section 4.2.

| Hyperparameter | Value |
|---|---|
| env_id | XLand-MiniGrid-R1-13x13 |
| benchmark_id | medium-1m |
| use_bf16 | True |
| pretrain_multitask | True |
| context_emb_dim | 16 |
| context_hidden_dim | 64 |
| context_dropout | 0.0 |
| obs_emb_dim | 16 |
| action_emb_dim | 16 |
| rnn_hidden_dim | 1024 |
| rnn_num_layers | 1 |
| head_hidden_dim | 256 |
| num_envs | 65536 |
| num_steps | 256 |
| update_epochs | 1 |
| num_minibatches | 64 |
| total_timesteps | 25,000,000,000 |
| optimizer | Adam |
| decay_lr | True |
| lr | 0.0005 |
| clip_eps | 0.2 |
| gamma | 0.995 |
| gae_lambda | 0.999 |
| ent_coef | 0.001 |
| vf_coef | 0.5 |
| max_grad_norm | 0.5 |
| eval_episodes | 256 |
| eval_seed | 42 |
| train_seed | 42 |

Table 10: PPO hyperparameters used in single-task fine-tuning from Section 4.2.

| Hyperparameter | Value |
| --- | --- |
| env_id | XLand-MiniGrid-R1-13x13 |
| benchmark_id | medium-1m |
| use_bf16 | True |
| pretrain_multitask | False |
| context_emb_dim | 16 |
| context_hidden_dim | 64 |
| context_dropout | 0.0 |
| obs_emb_dim | 16 |
| action_emb_dim | 16 |
| rnn_hidden_dim | 1024 |
| rnn_num_layers | 1 |
| head_hidden_dim | 256 |
| num_envs | 8192 |
| num_steps | 256 |
| update_epochs | 1 |
| num_minibatches | 8 |
| total_timesteps | 1,000,000,000 |
| optimizer | Adam |
| decay_lr | True |
| lr | 0.0005 |
| clip_eps | 0.2 |
| gamma | 0.995 |
| gae_lambda | 0.999 |
| ent_coef | 0.001 |
| vf_coef | 0.5 |
| max_grad_norm | 0.5 |
| eval_episodes | 256 |
| eval_seed | 42 |
| train_seed | 42 |

