# OpenReview forum: "XLand-100B: A Large-Scale Multi-Task Dataset for In-Context Reinforcement Learning"
_ICLR.cc/2025/Conference — ICLR 2025 Poster_

### Official Review · Reviewer_SXgP · 2024-10-26

**Soundness:** 2
**Presentation:** 2
**Contribution:** 2
**Rating:** 6
**Confidence:** 4

**Summary:**

In recent years, there have been some efforts on in-context reinforcement learning (ICRL). According to the authors, most ICRL research has been done on simple environments, and the lack of challenging benchmarks limits progress of the field. To address this, the paper proposes XLand-100B, a large scale dataset for in-context learning in gridworlds, comprising 30K tasks and 2.5B episodes.

**Strengths:**

The paper highlights an important limitation of current ICRL research, which is often focused on simple benchmarks such as gridworlds and small datasets. To this end, the authors generate a large-scale dataset and make it publicly available. The dataset contains episodes from 28K tasks, which is larger than other ICRL datasets. The authors make good design decisions in terms of their storage format (hdf5, compression levels, chunk size) and data collection strategy. As such, the dataset may be useful to the broader ICRL community.

**Weaknesses:**

While the paper highlights, that current ICRL research is focusing on simple environments, the gridworld environment they consider also seems simplistic. The generated dataset seems to be specific for a 5x5 single-room grid (not the multi-room in Figure 1? See questions). Generally, the environment seems similar to Dark-Room/Key-Door, and can be seen as a more advanced grid-world (with more than 2 objects) to test ICRL in toy environments. Therefore, it is unclear how well future findings on this dataset would transfer to other settings like robotics.

To the best of our understanding, there is a maximum of 9 rules in this environment, which correspond to reaching different objects in order. It would help the paper to provide a visual illustration of those rules for the reader to get an understanding of how diverse those tasks are (especially as it is a 5x5 grid). Due to those 9 rules, many different tasks can be produced. However, while there are 28K tasks in the generated dataset, it is unclear how much overlap or diversity there is between tasks. Therefore, it is possible that ICRL agents do not benefit from training on additional tasks. We suggest conducting an ablation study, in which agents are trained on 100/1000/10000/all tasks to evaluate the effect on ICRL abilities. This is currently missing.

While the authors release a large dataset with lots of tasks, from the perspective of a user it is unclear where to start. This is due to the lack of a clear benchmark split. The authors should clarify on what tasks to train on and what to evaluate on. This could be by number of pre-training tasks (as mentioned above) or number of rules etc. This point is important and needs significant revision and empirical evaluation of the considered methods.

In Figures 12 and 13 the learning curves (over 8K and 15K episodes) across all tasks are provided. The agents reach optimal performance quickly, only after a coupled hundred steps. Therefore, there is little learning progress in the remaining 14K updates. This may be a reason why AD and DPT do not exhibit meaningful in-context improvement during evaluation (Figure 7, 17, 20). Generally, there seems to be little in-context improvement by any ICRL method, which is different from previous ICRL works. This raises the question, whether the problem lies in the algorithms or the datasets, and should be discussed in the paper.

**Questions:**

- Can you clarify whether the dataset is collected on a 5x5 grid or on the grid visualized in Figure 1? If so, it is important to update Figure 1 to show the actual environment.
- In Figure 5, it seems that there is not much difference in learning performance from 1 to 7 rules, but a larger jump to 9 rules. Can you clarify why this is the case?
- Why does AD not exhibit in-context improvement in Figure 7 (only slightly for 1 rule)? Can you provide empirical evidence if this is a property of the data or the algorithm?
- In Figure 8, AD performance decreases with a larger context length. Why is that? Can you clarify how many episodes the context comprises?

---

> ### Author Response · Authors · 2024-11-20
>
> We thank the reviewer for the feedback and would like to discuss the weaknesses and questions outlined.
>
> > While the paper highlights, that current ICRL research is focusing on simple environments, the gridworld environment they consider also seems simplistic. The generated dataset seems to be specific for a 5x5 single-room grid (not the multi-room in Figure 1? See questions). Generally, the environment seems similar to Dark-Room/Key-Door, and can be seen as a more advanced grid-world (with more than 2 objects) to test ICRL in toy environments. Therefore, it is unclear how well future findings on this dataset would transfer to other settings like robotics.
>
> We understand that the grid-world nature of the benchmark might seem simplistic at a first glance, but we want to assure the reviewer that to solve it, the agent needs to thoroughly explore the environment and use the memory extensively. Firstly, let us clarify that 5x5 is the dimension of agent’s FOV, the environment itself is bigger: 9x9 in case of `-Trivial-20B`, 13x13 for `-100B`. Secondly, not only the benchmark proposed is equipped with substantially more objects, it is also in possession of a complex task generation strategy inherited from [1, 2] (which is explained answering the next reviewer’s question).
>
> Moreover, Meta-World [3], which is considered a suitable robotic benchmark for In-Context Reinforcement Learning (ICRL), offers only 50 tasks, all of which are conceptually straightforward (e.g., pushing, opening, inserting objects). In contrast, our dataset offer a more accessible playground with a large coverage of problems of varying complexity from very easy to very hard with the same underlying latent structure (binary task tree), similar to Alchemy [4] benchmark. We thus believe that out benchmark can produce interesting results that generalize to a broader class of tasks, similar to how it happened in the LLM filed, where many important general discoveries have been made on synthetic benchmarks testing specific abilities (e.g. induction heads, grokking, long context retrieval, etc).
>
> We also has made a demo of the environment, where one may try to solve the environment with # rules = 9 in different settings. There are three levels of difficulty: play with what the agent sees (its 5x5 observation), play with a bird’s eye view and play with a hint that can help solving the environment. The demo is available here: [http://play-xland.duckdns.org/](http://play-xland.duckdns.org/)

---

> ### Author Response · Authors · 2024-11-20
>
> > To the best of our understanding, there is a maximum of 9 rules in this environment, which correspond to reaching different objects in order. It would help the paper to provide a visual illustration of those rules for the reader to get an understanding of how diverse those tasks are (especially as it is a 5x5 grid). Due to those 9 rules, many different tasks can be produced. However, while there are 28K tasks in the generated dataset, it is unclear how much overlap or diversity there is between tasks. Therefore, it is possible that ICRL agents do not benefit from training on additional tasks.
>
> The rulesets are sampled from the original environment `medium-1m` benchmark [2], which contains 1 million different tasks. We sample 30 thousands tasks from 1M without overlap. The process of task generation is quite complex and is taken from [1, 2], but we want to explain it to highlight that tasks are in fact challenging for the agent. For a better understanding of the process of task generation, we briefly explain it here.
>
> Each task is represented as a tree where nodes are production rules and the root is the main goal. Task generation begins by sampling a goal and recursively adding production rules whose outputs serve as inputs to preceding levels, potentially forming a complete binary tree since rules have at most two arguments. At the start of an episode, only the objects from the leaf nodes are placed on the grid with randomized positions. The agent must trigger these rules in the correct sequence to obtain the necessary objects to achieve the goal, similar to task structures in benchmarks like Minecraft or Crafter. XLand-MiniGrid authors utilize ten colors and seven tile types for object sampling.
>
> To ensure each object appears only once as an input and once as an output in the main task tree, authors restrict input choices when sampling production rules by excluding already used objects. To enhance diversity, XLand-MiniGrid creators allow variable task depths and implement branch pruning; with some probability, a node is marked as a leaf, and its input objects are added to the initial objects. This approach generates tasks with varying branching factors or more sequential tasks with greater depth while using the same resource budget.
>
> To prevent the agent from employing brute-force methods, XLand’s authors introduce distractor rules and objects. Distractor objects are unused items not involved in any rules, and distractor production rules use objects from the main task tree but produce no useful outputs. These create dead ends that can render the game unsolvable, as each object is only present once. Consequently, the agent must experiment intelligently, remembering and avoiding these unhelpful rules to successfully complete the tasks.
>
> We added the visualization of aforementioned trees in Appendix L of our paper. For convenience reasons, we also share them by [by this link](https://postimg.cc/gallery/6NFQ17r). There are 3, 6 and 9 rules.
>
> > We suggest conducting an ablation study, in which agents are trained on 100/1000/10000/all tasks to evaluate the effect on ICRL abilities. This is currently missing.
>
> To clarify reviewer’s concerns, we conducted an additional experiment with variable number of goals. The results could be found by following [this anonymous link](https://i.postimg.cc/13R28fN0/exp-data-goals.png). Experiments are run with the sequence length of 2048. To make the number of gradient updates closer to the original experiments, we progressively increased the number of epochs for each smaller subset. Note that the return increases from 100 goals → 3k goals, indicating the importance of scaling the number of goals in the dataset. Running the AD for a single epoch on full goals dataset is clearly not enough, however, a large scale training is too computationally expensive for us at the moment and it is left for future research. We update the appendix with this experiment.
>
> > While the authors release a large dataset with lots of tasks, from the perspective of a user it is unclear where to start. This is due to the lack of a clear benchmark split. The authors should clarify on what tasks to train on and what to evaluate on. This could be by number of pre-training tasks (as mentioned above) or number of rules etc. This point is important and needs significant revision and empirical evaluation of the considered methods.
>
> In general, after pre-training on `-Trivial-20B` one should evaluate on tasks from XLand-MiniGrid [2] `trivial-1m` and for `-100B` on `medium-1m`. There is no need to split the tasks manually from the dataset since all of them are meant for training, thus we do not discuss it in the paper. Together with dataset, we plan to release the training code of AD and DPT that can be used as a starting point for future research. There we demonstrate how to download, pre-process and evaluate pre-trained models.

---

> ### Author Response · Authors · 2024-11-20
>
> > In Figures 12 and 13 the learning curves (over 8K and 15K episodes) across all tasks are provided. The agents reach optimal performance quickly, only after a coupled hundred steps. Therefore, there is little learning progress in the remaining 14K updates. This may be a reason why AD and DPT do not exhibit meaningful in-context improvement during evaluation (Figure 7, 17, 20). Generally, there seems to be little in-context improvement by any ICRL method, which is different from previous ICRL works. This raises the question, whether the problem lies in the algorithms or the datasets, and should be discussed in the paper.
>
> In case of AD, Laskin et al. [5] do not specify the length of policy improvement, nor any other details about the data collection apart from it should contain policy improvement operator (which our data does). More than that, they propose to subsample long trajectories, meaning that AD is capable of learning from faster algorithms. In case of DPT, Lee el al. [6] show that their method is able to leverage the datasets with high quantity of expert trajectories, meaning fast single-task policies should not pose a problem either. In Appendix H we discuss why DPT has failed in the POMDP setting and back up our hypothesis with experiments. In general, we believe that the performance of a strong ICRL algorithm should not depend on the data, when the criteria for the data collection procedure are met.
>
> > Can you clarify whether the dataset is collected on a 5x5 grid or on the grid visualized in Figure 1? If so, it is important to update Figure 1 to show the actual environment.
>
> The size of the grid depends on the dataset difficulty.  It is 9x9 in case of `-Trivial-20B`, 13x13 for `-100B` and the 5x5 are the dimensions of agent’s field of view. We added the renders of 9x9 and 13x13 environments to the appendix and left a link to them in the caption of Figure 1.
>
> > In Figure 5, it seems that there is not much difference in learning performance from 1 to 7 rules, but a larger jump to 9 rules. Can you clarify why this is the case?
>
> This is primary an artefact of our two-stage collection process with rule-conditioned pretraining with subsequent finetuning on masked rules. For simpler problems pretraining has the effect of improving the zero-shot performance (since it is possible to solve the task in just one or two episodes). For complex problems the role of pretraining is crucial, since it makes possible to solve them at all due to the more useful exploration prior. This exact effect can be seen in the plot: on simpler tasks, learning does not start at zero, but at some baseline reward above, which speeds up convergence on simpler tasks. At the same time, it does not happen on more complex tasks.
>
> > Why does AD not exhibit in-context improvement in Figure 7 (only slightly for 1 rule)? Can you provide empirical evidence if this is a property of the data or the algorithm?
>
> We speculate that AD demonstrates the in-context ability only on relatively easy tasks, while on the harder tasks its performance saturates quite quickly. It may be the case that AD, as opposed to DPT, has no convergence guarantees and for harder tasks with more exploration complexity the in-context ability of AD deteriorates significantly. We also would like to note that for # rules = 3 AD also demonstrates in-context ability, but is able to solve approximately 55% of evaluation tasks. Starting from # rules = 4, the algorithm is unable to solve harder tasks. We would like to point out that AD was previously tested on environments with rather simple tasks (Dark Room/Key-to-Door) [5, 6, 7] so there is no evidence of AD is able to perform in difficult environments.
>
> > In Figure 8, AD performance decreases with a larger context length. Why is that? Can you clarify how many episodes the context comprises?
>
> We believe that it becomes harder for transformer to attribute to certain actions with the increase in context size. It is possible that with more compute longer context would demonstrate better results for AD. On the other hand, it also highlights the imperfection of algorithm itself that needs large amount of compute. It is hard to estimate the accurate number of in-context episodes during evaluation due to the variable nature of episode termination. We report the number of episodes for the worst-case scenario when the time-limit is met in the table below. Consider these numbers as lower bounds.
>
> TL for `-100B` : 507
>
> | episodes/context length | 1024 | 2048 | 4096 |
> | --- | --- | --- | --- |
> | 100B | 2 | 4 | 8 |

---

> ### Author Response · Authors · 2024-11-20
>
> [1] Team, A. A., Bauer, J., Baumli, K., Baveja, S., Behbahani, F., Bhoopchand, A., ... & Zhang, L. (2023). Human-timescale adaptation in an open-ended task space. arXiv preprint arXiv:2301.07608.
>
> [2] Nikulin, A., Kurenkov, V., Zisman, I., Agarkov, A., Sinii, V., & Kolesnikov, S. (2023). XLand-minigrid: Scalable meta-reinforcement learning environments in JAX. *arXiv preprint arXiv:2312.12044*.
>
> [3] Yu, T., Quillen, D., He, Z., Julian, R., Hausman, K., Finn, C. & Levine, S.. (2020). Meta-World: A Benchmark and Evaluation for Multi-Task and Meta Reinforcement Learning.
>
> [4] Wang, J. X., King, M., Porcel, N., Kurth-Nelson, Z., Zhu, T., Deck, C., ... & Botvinick, M. (2021). Alchemy: A benchmark and analysis toolkit for meta-reinforcement learning agents. *arXiv preprint arXiv:2102.02926*.
>
> [5] Laskin, M., Wang, L., Oh, J., Parisotto, E., Spencer, S., Steigerwald, R., ... & Mnih, V. (2022). In-context reinforcement learning with algorithm distillation. *arXiv preprint arXiv:2210.14215*.
>
> [6] Grigsby, J., Fan, L., & Zhu, Y. (2023). Amago: Scalable in-context reinforcement learning for adaptive agents. arXiv preprint arXiv:2310.09971.
>
> [7] Dai, Z., Tomasi, F., & Ghiassian, S. (2024). In-context Exploration-Exploitation for Reinforcement Learning. arXiv preprint arXiv:2403.06826.

---

> ### Comment · Reviewer_SXgP · 2024-11-25
>
> We thank the authors for their detailed response and the clarifications. I still believe that the dataset may be useful to the broader ICRL community. Therefore, I decide to raise my score accordingly.

---

### Official Review · Reviewer_HFJ5 · 2024-11-02

**Soundness:** 3
**Presentation:** 4
**Contribution:** 4
**Rating:** 8
**Confidence:** 4

**Summary:**

This paper introduces XLand-100B, a large-scale data set for in-context reinforcement learning containing 100B transitions across 30,000 different tasks. The dataset includes complete learning histories of agents trained with PPO in the XLand-MiniGrid environment. The authors evaluate two common in-context RL algorithms on the dataset, showing the limitations of these methods with complex tasks. The work aims to democratize in-context RL research by providing a standardized, large-scale benchmark.

**Strengths:**

- The paper is well written.
- There do not seem to be any major issues with the method or the data set.
- Authors appear to have taken reproducibility seriously.
- The technical presentation of the data set, including relevant specifications, is comprehensive and clear.
- The work addresses an important gap that is hindering progress in an area of reinforcement learning of growing interest.

**Weaknesses:**

The paper has no critical weaknesses. However, for the sake of constructive academic discussion, I include some drawbacks below:

- The dataset exclusively uses PPO to generate learning histories, without theoretical or empirical justification for why PPO is especially suited for ICRL compared to other RL algorithms. Given the diversity of RL methods—each with distinct exploration strategies, convergence behaviors, and learning dynamics—relying solely on PPO risks biasing the dataset toward a specific style of learning history. This may inadvertently limit the dataset’s utility for evaluating ICRL methods that could benefit from more varied demonstration patterns.
- The dataset’s approach to ensuring high-quality demonstrations is primarily based on filtering out tasks with low final returns, which does not fully address whether all included histories are genuinely informative or relevant for ICRL.
- The use of approximate expert actions for labeling could compromise action fidelity, particularly for complex tasks, which may impact models that rely on high-quality expert demonstrations.

These weaknesses may affect the extent to which this benchmark is a good proxy for in-context RL. While benchmarks are very important for research progress, misaligned benchmarks can be actively counterproductive. On balance, I do not expect this to be a major problem, but it would be desirable if the authors strengthened the paper on this respect.

**Questions:**

- Could you clarify the rationale for choosing PPO as the sole algorithm for generating learning histories? How might the dataset be affected if other RL methods with different exploration and learning characteristics were incorporated?

- How do you ensure that filtered tasks with low returns are adequate proxies for high-quality demonstrations in ICRL?

- How might labeling inaccuracies impact the performance of models relying on these demonstrations?

---

> ### Author Response · Authors · 2024-11-20
>
> We greatly appreciate the reviewer’s high regard for our work and would like to address their questions and concerns.
>
> > Could you clarify the rationale for choosing PPO as the sole algorithm for generating learning histories? How might the dataset be affected if other RL methods with different exploration and learning characteristics were incorporated?
>
> We agree with the reviewer regarding the potential risks of using a sole algorithm for data collection. However, at small scale there is no evidence that the choice of data collection algorithm can affect the performance. As an example, in AD [1] authors use A2C to collect data in Dark Room and Key-to-Door environments, while authors of AD$\^{\varepsilon}$ [2] use Q-learning and the resulting performance does not differ. At the same time, it does not mean that this concerns will not appear on a large scale, so this might be a good topic for further research. The choice of PPO itself is determined by PPO's ability to scale to thousands of parallel environments, which allows to greatly accelerate the process of data collection using JAX.
>
> > How do you ensure that filtered tasks with low returns are adequate proxies for high-quality demonstrations in ICRL?
>
> We choose this approach following the data procedures from [1, 3], when it is stated that the learning histories must contain policy improvement for in-context ability to emerge. We believe such choice is vital when we train the model in the next-token prediction paradigm, but not in the classical RL settings with any kind of value-fucntion training, which could distinguish actions that lead to desired or undesired consequences. We surely consider the possibility of enlarging our dataset with low-quality demonstrations if there is such a demand in the future, since this task is relatively easy in terms of compute.
>
> > How might labeling inaccuracies impact the performance of models relying on these demonstrations?
>
> Lee et al. [3] show in their work that using PPO-trained experts for labeling target actions does not decrease DPT’s performance. However, we note that these experiments are performed on relatively small scale and further investigation is needed to empirically show the equivalence. On the other hand, Lin et al. [4] theoretically demonstrate that it is possible to estimate oracle actions with an algorithm, so we believe the inaccuracies should not greatly impact the final performance of DPT-like algorithms (in MDP setting at least).
>
> [1] Laskin, M., Wang, L., Oh, J., Parisotto, E., Spencer, S., Steigerwald, R., ... & Mnih, V. (2022). In-context reinforcement learning with algorithm distillation. *arXiv preprint arXiv:2210.14215*.
>
> [2] Zisman, I., Kurenkov, V., Nikulin, A., Sinii, V., & Kolesnikov, S. (2023). Emergence of In-Context Reinforcement Learning from Noise Distillation. arXiv preprint arXiv:2312.12275.
>
> [3] Lee, J., Xie, A., Pacchiano, A., Chandak, Y., Finn, C., Nachum, O., & Brunskill, E. (2024). Supervised pretraining can learn in-context reinforcement learning. Advances in Neural Information Processing Systems, 36.
>
> [4] Lin, L., Bai, Y., & Mei, S. (2023). Transformers as decision makers: Provable in-context reinforcement learning via supervised pretraining. *arXiv preprint arXiv:2310.08566*.

---

### Official Review · Reviewer_gcgV · 2024-11-02

**Soundness:** 3
**Presentation:** 4
**Contribution:** 3
**Rating:** 8
**Confidence:** 4

**Summary:**

This paper introduces new datasets “XLand-100B” and “XLand-Trivial-20B” intended for research on in-context reinforcement learning (ICRL). XLand-100B consists of 100B transitions spanning 30k different tasks from the XLand-MiniGrid environment.

The authors argue that research in ICRL has been hampered by the lack of large & diverse datasets that are compatible with ICRL, as well as by the significant compute requirements in this domain, and discuss how their contribution aims to make this research more accessible, in particular for researchers with limited resources. They emphasize that compared to prior work (NetHack, Procgen, GATO,...), their dataset is larger, more diverse, and better suited for ICRL research since it contains full learning histories instead of unordered or solely optimal/expert trajectories.

The data was collected by first pretraining a PPO multi-task policy on 65k tasks for 25B transitions and then finetuning it on a subset of 30k tasks. The dataset was only collected during the latter phase (for XLand-100B). The authors show that the pretrained policy performs well in the zero-shot setting, and that using it as a starting point for finetuning yields a very large performance boost compared to training a single-task policy from scratch (on hard tasks).

Finally, the authors benchmark Algorithm Distillation (AD) and Decision-Pretrained Transformer (DPT) on their dataset. They show promising results with AD, which displays an emergent ICL ability. However, they note that they were unable to show the same for DPT. The authors hypothesize that DPT performs poorly in POMDP settings as its lack of positional encoding prevents it from leveraging useful historical information from the current episode.

**Strengths:**

Overall, the paper makes a substantial contribution to in-context RL research (and RL research in general). The authors argue clearly what distinguishes their work from the many existing datasets and benchmarks, and as such, the design of the benchmark is very well-motivated. The paper is also well-written and the presentation is good.

Some strengths worth highlighting:

1. I particularly appreciate the level of detail regarding data collection, the dataset format, and other implementation/engineering methodology.
2. The authors made significant effort to make the benchmark easy to use (e.g. tuning the data compression to balance dataset size and sampling speed, and providing estimated optimal actions which are needed by methods such as DPT).
3. Including the lighter XLand-Trivial-20B dataset should make the benchmark significantly more accessible to researchers with limited resources.

I would be happy to increase the score, pending clarification on some of my concerns below.

**Weaknesses:**

I don’t believe there to be any significant weaknesses relating to the dataset (the main contribution) itself, apart from the limitations mentioned in the paper.

Regarding the AD & DPT baselines:

1. My understanding is that DPT, in contrast to AD, is unable to reason in POMDP settings because its transformer doesn’t include a positional encoding. Have you considered the modification of applying a positional encoding to just the transitions from the current (ongoing) episode? This wouldn’t significantly change the DPT formulation but would remove the limitation of not being able to use historical information from the current episode. The DPT experiment as it stands does not seem like a fair comparison to AD as its poor performance likely doesn’t come from the method itself but the observation space (lack of historical context).
2. It is stated that evaluation is performed for 512 episodes with a context length of k=1024/2048/4096. My understanding is that the context is initially empty and that at the end of evaluation, the context contains the most recent k transitions (potentially from multiple episodes). It’s unclear how often the context actually includes data from multiple episodes. What is the median episode length?
3. Have you performed experiments (with AD) where the context only contains data from the current episode? From Appendix H, it is clear that knowledge of the current episode history is essential for solving some of the partially-observable tasks (e.g. knowing whether you have already picked up a key). Including data from previous episodes could make it more difficult since the policy would need to distinguish whether a key pick-up happened in the current or previous episode.
4. Do you have any insight on why longer context lengths perform strictly worse with AD? Could it be related to the issue mentioned in my previous question (3.)?

Minor:

5. Figure 8 is a bit unclear as there is no label/legend for the colors. The 1024 context length could be confused with the 1024 unseen tasks used for evaluation.

**Questions:**

1. Why did you choose to use recurrent PPO with GRU for data generation over using a transformer (as in the two ICRL experiments)?
2. Evaluation is performed for 500 episodes and the score averaged across 1024 tasks. Am I correctly understanding this as performing 1024 separate single-task evaluations (each 500 episodes long) and then averaging scores across tasks? I.e., the transformer context never contains data from multiple tasks.

---

> ### Author Response · Authors · 2024-11-20
>
> We are glad the reviewer finds our work a substantial contribution and would like to answer the questions.
>
> > Have you considered the modification [of the DPT] of applying a positional encoding to just the transitions from the current (ongoing) episode?
>
> The process of training DPT is different from standard decoder next-token prediction. Instead of predicting next action for each state in the context (as in AD), in DPT they predict the same action for query state from each token in context, which makes it invariant to the order of tokens in the context (as authors note in [2]). That way, it is impossible to for DPT to attribute to the context as consecutive history, so the introduction of positional encodings probably will not help. On the other hand, what could be helpful is to pre-encode the consecutive interactions by any memory-model (essentially trying to recover true markovian state), say a simple RNN or LSTM and operate on such context. However, the main point of our benchmarks was to test the proposed algorithms as is and we left the further researchers to discover what works best to make DPT work in POMDP settings. We nevertheless wanted to draw attention to this, as neither in the original paper nor in the subsequent ones, when comparing DPT with AD, this limitation is not mentioned.
>
> > It’s unclear how often the context actually includes data from multiple episodes. What is the median episode length?
>
> The median episode length is shown in Table 2 of the main text. As reviewer requests, we also report the number of episodes that fit different context length on training and evaluation in the tables below. Note that it is hard to say what is the median episode length on evaluation, since it depends on the performance of the algorithm. We also report the worst case number of episodes, when the algorithm cannot solve the environment completely, so the episode runs until timelimit is met.
>
> **Train:**
>
> | episodes/context length | 1024 | 2048 | 4096 |
> | --- | --- | --- | --- |
> | Trivial-20B | 45 | 91 | 182 |
> | 100B | 17 | 35 | 70 |
>
> **Worst case eval:**
>
> TL for `-Trivial` :  243
>
> TL for `-100B` : 507
>
> | episodes/context length | 1024 | 2048 | 4096 |
> | --- | --- | --- | --- |
> | Trivial-20B | 4 | 8 | 16 |
> | 100B | 2 | 4 | 8 |
>
> > Have you performed experiments (with AD) where the context only contains data from the current episode?
>
> According to [1], the key property of AD that makes the in-context learning possible is its multi-episodic nature. However, you point is valid and somehow indicate that the episode has ended might be beneficial for the performance. At the same time, we want to note that tweaking existing algorithms to work better was not the main aim of our research and we would appreciate it if further research could significantly boost the metrics, especially on the hardest part of our dataset.
>
> > Do you have any insight on why longer context lengths perform strictly worse with AD? Could it be related to the issue mentioned in my previous question (3.)?
>
> It might be so that the absence of indication may introduce some confusion to the transformer attention and it would be interesting to investigate this. We believe that the main cause for underperformance on longer context is that the algorithms need more time to adapt for it, since the complexity is higher. We seek for the algorithms (and maybe backbone models) which can overcome this obstacle and demonstrate high performance despite the length of a context.
>
> > Figure 8 is a bit unclear as there is no label/legend for the colors. The 1024 context length could be confused with the 1024 unseen tasks used for evaluation.
>
> We fixed this concern and updated the figure in the new paper version, thank you for the suggestion.
>
> > Why did you choose to use recurrent PPO with GRU for data generation over using a transformer (as in the two ICRL experiments)?
>
>
> As we state in the paper, the capacity of PPO + GRU is enough to solve single-goal tasks. When task is fixed, as in pretraining, the main challenge is primarily in exploration not memory and transformer can not improve it. Also, note that the overhead of using GRU during data generation in the environment is significantly less than Transformer’s (due to it sequential nature which can not be parallelized).
>
> For AD, it is vital that the data consists of gradually improving policies [1], and PPO + GRU can provide these. DPT, at the same time, only needs enough exploration within its prior and the target expert actions. In ICLR we do not provide the algorithm which task it currently solves, so here the capacity requirements are much higher, therefore all the proposed methods use transformer backbone.

---

> ### Author Response · Authors · 2024-11-20
>
> > Evaluation is performed for 500 episodes and the score averaged across 1024 tasks. Am I correctly understanding this as performing 1024 separate single-task evaluations (each 500 episodes long) and then averaging scores across tasks? I.e., the transformer context never contains data from multiple tasks.
>
> We employ an evaluation scheme from [1] and [2], which indeed does not mix the goals inside one context. It is easier to think that we have a batch of size 1024, and in each batch we run its own environment with a specified goal for 500 full episodes. After the evaluation is over we average the returns across the tasks and across episodes, so for each episode we have an indication of in-context adaptation.
>
> [1] Laskin, M., Wang, L., Oh, J., Parisotto, E., Spencer, S., Steigerwald, R., ... & Mnih, V. (2022). In-context reinforcement learning with algorithm distillation. *arXiv preprint arXiv:2210.14215*.
>
> [2] Lee, J., Xie, A., Pacchiano, A., Chandak, Y., Finn, C., Nachum, O., & Brunskill, E. (2024). Supervised pretraining can learn in-context reinforcement learning. Advances in Neural Information Processing Systems, 36.

---

> > ### Comment · Reviewer_gcgV · 2024-11-25
> >
> > I appreciate the detailed responses to my questions as well as the minor update to the paper. I believe this new benchmark is a valuable contribution to ICLR and I raised my score accordingly.

---

### Official Review · Reviewer_4NsP · 2024-11-03

**Soundness:** 3
**Presentation:** 3
**Contribution:** 3
**Rating:** 6
**Confidence:** 3

**Summary:**

The paper introduces a dataset, XLand-100B, for training and testing in-context RL algorithms. In-context RL is the problem of predicting how to act in a task, given trajectories from the task, without any updates to the model weights. There is prior research in this field, but future progress is hindered by lack of open-source datasets/benchmarks where algorithms can be tested. The paper introduces a large scale dataset containing 100B transitions, 2.5B episodes from 30,000 (in effect, around 29,000 after filtering). The paper also tests two current algorithms, AD and DPT, to show that more progress/research is needed in the field.

**Strengths:**

1. The paper is a dataset paper, and I think it clearly motivates the need for such a dataset in the open-source community. In that way, it is a strong work, since the release of a dataset/hard enough benchmark can truly boost the research productivity in this important field. The release of the ImageNet benchmark accelerated the pace at which computer vision grew as a field, and a well-designed benchmark can do that for in-context RL as well, hence I support the acceptance of this paper.
2. The paper is nicely structured and well-written, giving insights and reasoning to the community.
3. As the authors claim, researchers in this field often need to generate their own data. This results in lack of reproducibility and wastage of computational resources. This paper can help mitigate some of those challenges.
4. The authors have spent significant effort in trying to compress the dataset while maintaining throughput for loading the dataset from the compressed version. This effort is highly appreciated!

**Weaknesses:**

1. Despite the enormous size of the dataset, it only contains one type of environment, based on the Mini-Grid set of tasks. While this is a starting point for research in this direction, adding tasks from more realistic domains, like robotics/other tasks that strictly require RL, would be appreciated. This is my main concern for not giving a higher score.
2. Adding more visualizations of the tasks, trajectories, etc in the appendix might be helpful to understand the precise nature of the tasks, what does # rule 3 → # rule 9 actually mean for the hardness of the task, etc.
3. The idea of collecting such a dataset is not novel, despite the importance of doing such work that can empower future novel research in this direction.

**Questions:**

Line 350

> In contrast to AD, which predicts next actions from the trajectory itself, DPT-like methods require access to optimal actions on each transition for prediction. However, for the most nontrivial or real-world problems, obtaining true optimal actions in large numbers is unlikely to be possible.

I am uncertain about this line. As shown by [1], one can use a fully finetuned RL policy for a task, that achieves good performance, to collect “expert” data. Why is that not the case for this paper’s tasks?

# References

[1] D4RL: Datasets for Deep Data-Driven Reinforcement Learning, https://arxiv.org/abs/2004.07219

---

> ### Author Response · Authors · 2024-11-20
>
> We are grateful to the reviewer for the valuable feedback and would like to address the questions in it.
>
> > Despite the enormous size of the dataset, it only contains one type of environment. […] While this is a starting point for research in this direction, adding tasks from more realistic domains, like robotics/other tasks that strictly require RL, would be appreciated
>
> We agree with the reviewer that enhancing the dataset with more complex data would benefit the dataset and the research in general. However, we want to highlight that in In-Context RL (ICRL) settings our dataset might be considered as one of the most complicated in terms of task diversity and exploration complexity.
>
> For example, in Meta-World [1], the suitable robotic benchmark for ICRL, there are only 50 tasks in total and the conceptual difficulty is quite low (push, open, insert, etc.) and do not vary significantly. In contrast, our dataset offer a more accessible playground with a large coverage of problems of varying complexity from very easy to very hard with the same underlying latent structure (binary task tree), similar to Alchemy [2] benchmark.
>
> In addition, given how much slower other benchmarks are (Meta-World, Alchemy, etc), it would unfortunately take a lot more resources to build a dataset comparable to ours. We, like most of the academic research labs, do not have such resources. That's what motivated us in the first place.
>
> We nevertheless believe that even such “synthetic” benchmarks can produce interesting results that generalize to a broader class of tasks, similar to how it happened in the LLM filed, where many important general discoveries have been made on synthetic benchmarks testing specific abilities (e.g. induction heads, grokking, long context retrieval, etc).
>
> > Adding more visualizations of the tasks, trajectories, etc in the appendix might be helpful to understand the precise nature of the tasks, what does # rule 3 → # rule 9 actually mean for the hardness of the task, etc.
>
> As the reviewer requested, we added a visualization of tasks with different complexities to the appendix (see Appendix M in the updated version). Please note that it is impossible to show the trajectories of an agent solving the environment, since the typical trajectory consists of around 500 steps for 9 rules. Instead, we show the task tree which the agent should explore to successfully solve an environment. For convenience, we also attach the plots here. The trees are also available [by this link](https://postimg.cc/gallery/6NFQ17r). There are 3, 6 and 9 rules.
>
> > The idea of collecting such a dataset is not novel, despite the importance of doing such work that can empower future novel research in this direction.
>
> It is true that there are plenty of RL datasets exists, however we think that we are the first who present a unified benchmark for ICRL. Recent works in this field use similar environments [2, 3, 4, 5], but with different environment parameters like size of the environment. These differences in training data can substantially affect the results of algorithms making comparisons flawed. While our dataset standardizes the data and evaluation process in ICRL, making the comparison of different algorithms reliable.
>
> > […] one can use a fully finetuned RL policy for a task, that achieves good performance, to collect “expert” data. Why is that not the case for this paper’s tasks?
>
> In the excerpt on line 350 we highlight the inability to collect true optimal actions in real-world environments. However, in the next sentence we say that there exists the provable method to substitute optimal actions with the actions from an expert policy [6]. We indeed use the expert policies for collecting target actions for DPT [5]. We store them in our dataset with the key `expert_actions`, more information on what else we store and how to access it is located in Appendix B.

---

> ### Author Response · Authors · 2024-11-20
>
> [1] Yu, T., Quillen, D., He, Z., Julian, R., Hausman, K., Finn, C. & Levine, S.. (2020). Meta-World: A Benchmark and Evaluation for Multi-Task and Meta Reinforcement Learning. <i>Proceedings of the Conference on Robot Learning</i>, in <i>Proceedings of Machine Learning Research</i> 100:1094-1100 Available from https://proceedings.mlr.press/v100/yu20a.html.
>
> [2] Wang, J. X., King, M., Porcel, N., Kurth-Nelson, Z., Zhu, T., Deck, C., ... & Botvinick, M. (2021). Alchemy: A benchmark and analysis toolkit for meta-reinforcement learning agents. *arXiv preprint arXiv:2102.02926*.
>
> [3] Schmied, T., Paischer, F., Patil, V., Hofmarcher, M., Pascanu, R., & Hochreiter, S. (2024). Retrieval-Augmented Decision Transformer: External Memory for In-context RL. arXiv preprint arXiv:2410.07071.
>
> [4] Laskin, M., Wang, L., Oh, J., Parisotto, E., Spencer, S., Steigerwald, R., ... & Mnih, V. (2022). In-context reinforcement learning with algorithm distillation. *arXiv preprint arXiv:2210.14215*.
>
> [5] Lee, J., Xie, A., Pacchiano, A., Chandak, Y., Finn, C., Nachum, O., & Brunskill, E. (2024). Supervised pretraining can learn in-context reinforcement learning. Advances in Neural Information Processing Systems, 36.
>
> [6] Lin, L., Bai, Y., & Mei, S. (2023). Transformers as decision makers: Provable in-context reinforcement learning via supervised pretraining. arXiv preprint arXiv:2310.08566.

---

> > ### Comment · Reviewer_4NsP · 2024-11-20
> >
> > I thank the authors for their thoughtful rebuttal. My concerns have been mostly addressed, but I would like to keep the same score.

---

### Meta-Review · Area_Chair_Gemw · 2024-12-14

**Metareview:**

The authors present a new dataset for training and testing in-context RL algorithms that goes beyond existing tasks in simple environments and on small-scale datasets. They also provide results from running common baselines, showing that existing methods struggle to generalize to novel and diverse tasks.

Reviewers found the paper well-written and and containing a significant amount of detail about how the dataset is collected. The benchmark seems easy to use and made efforts to ensure reproducibility. However, the reviewers also raised concerns that the contribution is limited in the sense that this benchmark is built on only one type of environment, Mini-Grid and all baselines implemented using the same underlying RL algorithm, PPO. Finally, there are concerns that no ICRL method seems to perform well on the benchmark, raising the question if the issue is with ICRL methods or the benchmark itself.

Because reviewers agree that the benchmark is thoughtfully designed, well-motivated, and timely for research in ICRL, I vote to accept this paper.

**Additional Comments On Reviewer Discussion:**

Most reviewers found the authors addressed their concerns with minimal changes needed to the paper, resulting in raised or maintained scores.

---

### Decision · Program_Chairs · 2025-01-22

Accept (Poster)